# Effect of Type 2 Diabetes and Impaired Glucose Tolerance on Digestive Enzymes and Glucose Absorption in the Small Intestine of Young Rats

**DOI:** 10.3390/nu14020385

**Published:** 2022-01-17

**Authors:** Lyudmila V. Gromova, Alexandr S. Polozov, Elizaveta V. Savochkina, Anna S. Alekseeva, Yulia V. Dmitrieva, Oleg V. Kornyushin, Andrey A. Gruzdkov

**Affiliations:** 1Pavlov Institute of Physiology, Russian Academy of Sciences, 6 Makarova emb., 199034 Saint-Petersburg, Russia; gromovalv@infran.ru (L.V.G.); polozovas@infran.ru (A.S.P.); savochkinaev@infran.ru (E.V.S.); alekseevaas@infran.ru (A.S.A.); dmitrievayv@infran.ru (Y.V.D.); 2Almazov National Medical Research Center, Ministry of Health of the Russian Federation, 2 Akkuratova Str., 197341 Saint-Petersburg, Russia; o.kornyushin@gmail.com

**Keywords:** type 2 diabetes, digestive enzymes, glucose absorption, transporters SGLT1, GLUT2

## Abstract

The reactions of intestinal functional parameters to type 2 diabetes at a young age remain unclear. The study aimed to assess changes in the activity of intestinal enzymes, glucose absorption, transporter content (SGLT1, GLUT2) and intestinal structure in young Wistar rats with type 2 diabetes (T2D) and impaired glucose tolerance (IGT). To induce these conditions in the T2D (*n* = 4) and IGT (*n* = 6) rats, we used a high-fat diet and a low dose of streptozotocin. Rats fed a high-fat diet (HFD) (*n* = 6) or a standard diet (SCD) (*n* = 6) were used as controls. The results showed that in T2D rats, the ability of the small intestine to absorb glucose was higher in comparison to HFD rats (*p* < 0.05). This was accompanied by a tendency towards an increase in the number of enterocytes on the villi of the small intestine in the absence of changes in the content of SGLT1 and GLUT2 in the brush border membrane of the enterocytes. T2D rats also showed lower maltase and alkaline phosphatase (AP) activity in the jejunal mucosa compared to the IGT rats (*p* < 0.05) and lower AP activity in the colon contents compared to the HFD (*p* < 0.05) and IGT (*p* < 0.05) rats. Thus, this study provides insights into the adaptation of the functional and structural parameters of the small intestine in the development of type 2 diabetes and impaired glucose tolerance in young representatives.

## 1. Introduction

In recent decades, many countries have seen an increase in the incidence of type 2 diabetes (T2D) among children and adolescents, accompanied by an increase in obesity in this age group [1,2]. In young people, as in adults with T2D, this disease is initially characterized by hyperglycemia arising from insulin resistance and impaired insulin secretion [1,2,3,4,5]. However, unlike older people, younger people with type 2 diabetes tend to suffer more from insulin deficiency, whereas older people tend to be more insulin resistant [1,6].

With regard to previous studies of the effect of type 2 diabetes on the gastrointestinal tract (GIT) in childhood and adolescence, it should be noted that such data are relatively few, and they are presented mainly in models of type 2 diabetes induced in young animals (for example, rats under 6 months) [7,8,9].

The results accumulated over the past 50 years (including studies on young animals) have shown that the main intestinal glucosidases (for example, glucoamylase, maltase) and glucose transporters in brush border membrane enterocytes (SGLT1, which actively transports glucose, and GLUT2, a transporter that operates via facilitated diffusion) are crucial for the formation of hyperglycemia in T1D and T2D [4,10,11,12,13].

This allows us to consider these factors as key targets for glycemic control in T2D. At the same time, strong evidence has been obtained showing that the SGLT1 transporter plays a predominant role in the absorption of glucose in the small intestine under normal conditions, and its contribution to this process increases markedly in type 1 and 2 diabetes [4,13,14,15,16,17]. As to the response of the GLUT2 transporter to type 2 diabetes, the available data are contradictory [18,19,20,21].

A number of works report that metabolic diseases can occur as the result of the metabolic inflammation caused by the increased intestinal transfer of bacterial toxin lipopolysaccharide, which is a part of the membrane of Gram-negative bacteria [22,23]. At the same time, it was found that intestinal alkaline phosphatase (AP) might detoxify this lipopolysaccharide, thereby confirming the crucial role of this enzyme not only in digestion, but also in maintaining intestinal homeostasis [23,24,25]. Indeed, a decrease in the AP level in the feces of animals and humans, found for metabolic syndrome and type 1 and 2 diabetes mellitus, may indicate that there is a high risk of developing an inflammatory process in the colon under these conditions [26,27,28]. In this regard, it is of great interest to analyze the AP response in the content of the colon through a model of type 2 diabetes in young animals.

Currently, there is relatively little information on the response of another important brush border enzyme in enterocytes, aminopeptidase N (APN), to type 2 diabetes in both adults and young individuals. According to the latest data from the literature, this enzyme, as with AP, is multifunctional. In addition to participating in digestion (the final stages of protein hydrolysis), it also performs other important functions: it participates in the transport of cholesterol in enterocytes, in immune responses and in the presentation of the antigen [23,29,30,31,32].

The aim of this work was to evaluate the reaction of the glucose absorption system in the small intestine in young rats during the development of type 2 diabetes and impaired glucose tolerance. Along with this, we investigated changes in the expression of glucose transporters SGLT1 and GLUT2 in the brush border membrane of jejunal enterocytes, the activities of a number of important intestinal enzymes (glucoamylase, maltase, alkaline phosphatase and aminopeptidase N), as well as the structural parameters of the small intestine at the end of the experimental period (9 weeks after the induction of these states).

## 2. Materials and Methods

### 2.1. Animals and the Type 2 Diabetes Model

We used Wistar rats (male, 4 weeks old, body weight 90–100 g) obtained from the Center for Collective Use in the I.P. Pavlov Institute of Physiology, RAS. The experiments were performed in full compliance with the European Council Directive (86/609/EEC) on the observance of ethical principles when working with animals, and approved by the Ethics Committee of the Pavlov Institute of Physiology, Russian Academy of Sciences.

A high-fat diet/low-dose streptozotocin model was used as a type 2 diabetes model in the present study [6]. Before inducing type 2 diabetes, animals were kept for 8 weeks on a high-fat diet (HFD) (20% protein, 22% fat, 25.4% starch and 1.83% sucrose, as a % of the wet weight taken as 100). The animals were then injected with streptozotocin (dose 30 mg/kg intraperitoneally), which, according to the methodology used, was the last step in the induction of type 2 diabetes. Under the same conditions, control animals fed a high-fat diet for 8 weeks were injected with citrate buffer (pH 4.5) (a solvent of streptozotocin). Throughout the experimental period, the animals continued to adhere to the same high-fat diet.

Additionally, there was a second control group of healthy rats kept on a standard chow diet (SCD) (20.0% protein, 6.0% fat, 37.18% starch and 2.82% sucrose, as a % of the wet weight taken as 100) throughout the experimental period. These control rats were used at the end of the experimental period mainly to compare their performance in the OGTT and ITT tests with that of the control rats fed a high-fat diet.

### 2.2. Experimental Design

The design of the experiment is presented in Figure 1.

### 2.3. Intestinal Glucose Absorption

The maximal capability of the rat small intestine to absorb glucose was assessed by the average rate of free (voluntary) consumption of a concentrated (20%) glucose solution by the animals after their fasting for 18–20 h, according to the novel technique previously published [33,34].

In brief, the results of our earlier experiments and their mathematical analysis have shown that this rate may serve as a reliable, objective, quantitative criterion by which to evaluate the maximal capability of the rat small intestine to absorb glucose under natural conditions [33,34]. This technique is based on the fact that rats, after fasting for 18–20 h, consume concentrated glucose solution at an almost constant average rate for several hours that reflects the maximal rate of its absorption in the intestine. This is due to the existence of so called “ileal brake”—the mechanism of control over the stomach emptying and intestinal motility in accordance with the maximally capable rate of nutrient absorption in the small intestine [35].

During the experiments, each rat (preliminarily fasted for 18–20 h) was placed into an individual cage, with a size of 14 × 21 × 11 cm, and equipped with two graduated nipple drinkers: one with water and another with glucose solution (200 g/L).

At first, on each of the rats, several tentative experiments were carried out (at 3–4-day intervals), during which the temporal dynamics of glucose solution consumption were examined. To do this, 15, 30, 45 and 60 min after the onset of the experiment and then every 30 min for 5–6 h, volumes of the solution consumed by each animal were measured. Animals that did not drink glucose solution in any of the tentative experiments were expelled from further experiments. Then, by linear regression using Origin 7.0 (OriginLab Corp., Northampton, MA, USA), the mean volume-normalized uptake rate of glucose solution (µL/min) was determined within the time interval that corresponded to a relative constancy of this parameter (from 60–90 to 330–360 min since the onset of the experiment).

According to the design, in preliminary experiments (before inducing type 2 diabetes), the rates of consumption of the glucose solution (in μL/min) were estimated for each of the animals. At first, the control (the HFD group, *n* = 6) and experimental (*n* = 20) groups were formed, based on these experiments, with similar average values of these rates.

The assessment of consumption of the glucose solution by rats in all groups of animals was carried out 4 times within 12 weeks after the first injection of streptozotocin (the age of the animals was then about 23 weeks, which corresponds to a young adult age).

### 2.4. Oral Glucose Tolerance Test (OGTT) and Insulin Tolerance Test (ITT)

During the OGTT, the blood in rats was collected from the tail vein after 18 h of fasting (point 0), and then 30, 60, 90 and 120 min after the oral administration of D-glucose solution (30%) at a dose of 2 g glucose/kg of animal body weight. The ITT was carried out according to the following scheme: at 10 a.m., after 18 h of fasting, the rats were injected intraperitoneally with insulin Humalog (0.75 IU/kg). Blood was collected before (point 0) and 30, 60 and 90 min after the insulin injection. The blood glucose levels were measured with a glucometer Accu-Chek *Performa* (Mannheim, Germany). The area under the curve for glucose in the OGTT (AUC) was determined using the trapezoidal rule.

According to the results of the final OGTT test (at the end of the experimental period), the animals of the different groups had the following characteristics: (1) the T2D group—the blood glucose level after 2 h of the OGTT ≥ 11.1 mmol/L, and the AUC was 22.8–24.6 mmol/L·h, *n* = 4; (2) the IGT group—the blood glucose level after 2 h of the OGTT ≥ 7.8 and < 11.1 mmol/L, and the AUC was 17.9–19.0 mmol/L·h, *n* = 6; (3) the HFD control group—the blood glucose level after 2 h of the OGTT < 7.8 mmol/L, and the AUC was 13.3–13.9 mmol/L·h, *n* = 6); and (4) the SCD control group (SCD)—the blood glucose level after 2 h of the OGTT < 7.8 mmol/L, and the AUC was 13.3–14.7 mmol/L·h, *n* = 6.

### 2.5. Collection of Intestinal Tissues from Various Areas and Contents of the Colon

In the final stage of the experiments, after a high glucose load on the small intestine (during free consumption by rats of 20% glucose solution for 4 h) a decapitation of the animals was performed. Then, samples of tissues of the middle part of the jejunum and ileum were taken for the analysis of the morphometric parameters and the content of the SGLT1 and GLUT2 transporters in the enterocytes using the immunohistochemical method. In addition, samples of the mucosa from various parts of the intestine and chyme from the colon were taken to determine the activity of the membrane digestive enzymes in them.

### 2.6. The Count of Enterocytes and Goblet Cells

Morphometric analysis was performed on well-oriented sections of the mucosa of the small intestine using a light microscope with an eyepiece equipped with an eyepiece micrometer. The number of enterocytes per longitudinal section of the villi was determined (at least 10 sections of the villi were analyzed for each section of the small intestine), as was the number of goblet cells in the epithelium of the villi.

### 2.7. Determination of the Content of Glucose Transporters in Enterocytes

The contents of the transporters of glucose were determined in the brush border of the enterocytes using the immunohistochemical method, and confocal microscopy was used as previously described [36]. When the rats were sacrificed, pieces of the jejunum were fixed in the solution of 2% formaldehyde in PBS for 1 h. Then, the samples were impregnated with 30% sucrose and were frozen with liquid nitrogen. The samples were stored at −30 °C before use. Sections with a 7–10 μm thickness were obtained using a Leica cryostat (Nussloch, Germany).

The sugar transporters were revealed via indirect immunofluorescence. Polyclonal goat antibodies to SGLT1 and rabbit antibodies to GLUT2 (Santa Cruz Biotechnology Inc., Dallas, TX, USA) diluted at 1:1000 with 0.5% BSA/0.3% Triton X_100/PBS were used as the primary antibodies. Anti-goat immunoglobulins conjugated to Alexa Fluor 633 fluorescent dye and anti-rabbit immunoglobulins conjugated to Alexa Fluor 488 fluorescent dye (Molecular Probes, Eugene, OR, USA) diluted at 1:500 with 0.5% BSA/0.3% Triton X 100/PBS were used as the secondary antibodies. The sections were embedded in a polyvinyl alcohol mounting medium with DABCO (Fluka, Buchs, Switzerland) under cover glass.

The finished sections were examined using a Leica TCS SL confocal microscope (Wetzlar, Germany). The contents of the glucose transporters were assessed semi-quantitatively using the Image J software by measuring the average intensity of the immunofluorescent fluorescence (in relative units) in the strips covering the brush border of the enterocytes located along the villus from the upper to the middle thirds (at least 20 sections of the villi were analyzed for each section of the small intestine).

### 2.8. Determination of the Activity of Intestinal Digestive Enzymes

The activities of the intestinal enzymes were determined in the homogenates of the mucosa and chyme using the standard biochemical techniques. The regents used are also presented here (Appendix A). The activity of glucoamylase (GA; E.C. 3.2.1.3) was determined with 12 g/L of starch as a substrate and the liberated glucose was measured with the Tris-glucose oxidase–peroxidase reagent [37], maltase (M; E.C. 3.2.1.20) was determined by the method of Dahlqvist [38], aminopeptidase N (APN; E.C. 3.4.11.2) was determined by the method of Farr et al. [39] and alkaline phosphatase (AP; E.C. 3.1.3.1) was determined using 0.6 mMol/l p-nitrophenylphosphate as a substrate and Ringer solution (pH 7.4) as buffer [37].

### 2.9. Statistical Analysis

Results are represented as the mean ± standard error of the mean (SEM). The significance of differences between the groups in terms of the studied parameters was assessed using the two-way ANOVA in the Origin 7.0 software (OriginLab Corp., Northampton, MA, USA), or the unpaired Student’s *t*-test, or in some cases the non-parametric method—the paired Wilcoxon test. Linear regression for variables and linear correlations between variables were calculated using Origin 7.0 software.Differences were considered to be statistically significant with *p* values < 0.05.

## 3. Results

### 3.1. Rat Body Weight

During the first 40 days after the administration of streptozotocin (Str), the increase in the body weight of rats in the IGT group did not significantly differ in magnitude from the control (HFD group) (1.22 ± 0.10 and 0.81 ± 0.03 g/day, respectively) (Figure 2A). Over the next 40 days, this indicator in the IGT group tended to decrease in comparison with the HFD group (Figure 2B). At the same time, in group T2D, the increase in the body weight of rats during the first 40 days was significantly lower than in the HFD and IGT groups (by 62%, *p* < 0.05 and 49%, *p* < 0.02, respectively) (Figure 2A). However, over the next 40 days (Figure 2B), the value of this indicator doubled compared to the initial period (*p* < 0.0027), and as a result, it became 59% higher than in the IGT group (*p* < 0.0027) and was close to that in the HFD group.

### 3.2. The Oral Glucose Tolerance and the Insulin Tolerance Tests in Rats

Figure 3 shows the results of the OGTT at 2 and 10 weeks and the ITT at 11 weeks after the first administration of Str or its diluent to rats from the T2D, IGT, HFD and SCD groups. Compared to the OGTT 2 weeks after Str administration, in the T2D group, the blood glucose concentration in the OGTT 10 weeks after Str administration had increased mainly by 120 min of this test (*p* < 0.05) (Figure 3A,B). At the same time, in the ITG group, the blood glucose concentration during the OGTT 10 weeks after the administration of Str significantly increased for all periods of the test compared to their values 2 weeks after the administration of Str (*p* < 0.05).

As seen in Figure 3B, the long-term high-fat diet (HFD) did not affect the blood glucose levels of animals who were measured after 18 h of fasting, as well as the levels of this indicator at 30, 60, 90 and 120 min after the oral administration of the glucose solution (at a dose of 2 g glucose/kg body weight). Even after 120 min, glucose levels in the HFD group were only slightly higher than in the SCD group (*p* > 0.05). In rats of the IGT and T2D groups, the blood glucose level at all periods after the oral administration of glucose was significantly higher than in the SCD and HFD groups (*p* < 0.05). In addition, in the T2D group, blood glucose levels were higher compared to the SCD group (*p* < 0.05).

During ITT (Figure 3C), 30 and 60 min after insulin administration, the blood glucose level of rats in the SCD group did not change, while in the HFD, IGT and T2D groups, this parameter was noticeably reduced compared to the initial level (0 min) (*p* < 0.05 paired Wilcoxon test). The most pronounced decrease in the blood glucose level took place in rats of the HFD group 30 min after the administration of insulin (*p* < 0.05 compared with the IGT group). These data confirm the presence of insulin resistance in the HFD, IGT and T2D groups, which is more pronounced in the IGT and T2D groups.

### 3.3. The Number of Enterocytes and Goblet Cells on the Villi of the Jejunum and Ileum

The long-term high-fat diet in the HFD group in the period from 4 weeks after birth to about 23 weeks of age did not significantly affect the cell count in the villi of the jejunum and ileum (Figure 4).

In the T2D group, there was a slight increase in the number of enterocytes on the villi in the jejunum and the ileum compared to the control HFD group (*p* < 0.05 in the ileum) (Figure 4). However, in the IGT group, the number of enterocytes on the villi was markedly reduced in both the jejunum and the ileum compared with other groups, but especially compared with the T2D group (by 23.2%, *p* < 0.02 and 33.1%, *p* < 0.0027, in the jejunum and in the ileum, respectively).

Under the influence of a long-term high-fat diet in the HFD group, there was a decrease in the number of goblet cells on the villi of the jejunum, but not on the ileum, compared with the control group SCD, which received a standard chow diet (*p* < 0.02) (Figure 4). In the T2D group, the number of goblet cells on the villi of the jejunum and ileum did not differ significantly from their values in the SCD and HFD controls. At the same time, in the IGT group, the number of goblet cells on the villi was noticeably lower in the jejunum compared with the SCD group (by 34.4%, *p* < 0.0027) and T2D group (by 26.5%, *p* < 0.01) and in the ileum compared with the T2D group (by 25.2%, *p* < 0.05).

### 3.4. Glucose Absorption, SGLT1 and GLUT2 Transporters

At 24 days after Str administration in the IGT and T2D groups, the capability to absorb glucose in the rat small intestine was only slightly higher than in the HFD control group (Figure 5). Later, at 42 and 66 days after the administration of Str, the increase in the rate of glucose absorption in the T2D group was higher than in the HFD and IGT groups (0.420 ± 0.066 versus 0.251 ± 0.016 and 0.190 ± 0.089 µmol/min per day, respectively, *p* < 0.05). As a result, the maximal rate of glucose absorption in the small intestine in the T2D group after 66 days became significantly higher than that in the HFD group (by 25%, *p* < 0.05).

In this study, the content of glucose transporters SGLT1 and GLUT2 in the brush border membrane of enterocytes in the T2D and HFD groups was determined at the end of the last experiment following the free consumption of glucose solution by rats (i.e., under the conditions close to natural). This point is especially important when analyzing the content of the GLUT2 transporter, which is able to quickly leave the brush border membrane of enterocytes with a decrease in glucose concentration in the lumen of the small intestine [18]. In addition, we have chosen for analysis the enterocytes from the area located in the middle part of the jejunum, that is, where the expression of these transporters is usually most pronounced.

Figure 6 shows microscopic images of the results of the immunohistochemical analysis of transporters SGLT1 and GLUT2 in enterocytes of the small intestine of rats.

A quantitative assessment of the intensity of the immunoluminescence of the GLUT2 transporter labels in the brush border membrane of the jejunal enterocytes did not reveal noticeable differences between the groups T2D and HFD (17.21 ± 2.80 and 19.09 ± 7.46 arb. units, respectively) (Figure 7). With regard to the average intensity of the immunoluminescence of the SGLT1 transporter labels in the brush border membranes of enterocytes in groups T2D and HFD, there was also no noticeable difference (12.35 ± 5.19 and 11.52 ± 3.59 arb. units).

### 3.5. Intestinal Enzyme Activities

*Glucoamylase*. A long-term high-fat diet in the HFD control group did not significantly alter the glucoamylase activity in the mucosa of various intestinal regions compared to that in the SCD control group (Figure 8A). At the same time, in the IGT and T2D groups, a decrease in glucoamylase activity in the duodenum was observed (in the IGT group: by 34.4%, *p* < 0.05 and 39.8%, *p* < 0.0027, and in the T2D group: by 25.0%, *p* < 0.05 and 33.2%, *p* < 0.0027, compared to the HFD and SCD groups, respectively).

*Maltase*. Under the influence of a long-term high-fat diet in the HFD and IGT groups, a tendency to increase the maltase activity in the jejunum and ileum compared to the SCD control group was observed. (Figure 8B). However, in the T2D group, there was a decrease in maltase activity in the mucosa of the distal part of the jejunum compared with the IGT group (by 52.9%, *p* < 0.05).

*Alkaline phosphatase.* The long-term high-fat diet reduced alkaline phosphatase (AP) activity in the duodenal mucosa in rats of the HFD group compared to the SCD group (by 46%, *p* < 0.05) (Figure 9A). At the same time, in rats in the IGT group, the activity of this enzyme in the mucosa was increased in the duodenum (by 2.0 times, compared with the HFD group), in the jejunum (in its distal part) (by 66.0, 74.3 and 53.0% compared with the SCD, HFD and T2D groups, respectively, *p* < 0.05), in the ileum (by 59% compared to the SCD group, *p* < 0.05,) and in the colon (2.0 times compared to the HFD group, *p* < 0.05). In group T2D, the activity of AP in the mucosa was increased in the duodenum and colon compared with the HFD group (by 2.2 and 1.8 times, respectively, *p* < 0.05).

*Aminopeptidase N*. The activity of aminopeptidase N (APN) in the mucosa of different parts of the intestine did not change significantly in rats of the HFD control group and in the T2D group compared to the SCD control group (Figure 9B). However, in the IGT group, the activity of this enzyme in the mucosa was reduced by 17.1% in the ileum compared to the SCD group (*p* < 0.05) and by 34.2% in the colon compared to the HFD group (*p* < 0.05).

### 3.6. AP and APN Activities in Colon Content

In rats of the HFD and IGT groups, the level of AP activity in the colon chyme was higher than in the SCD control group (>2 times, *p* < 0.05) (Figure 10A). In the T2D group, AP activity in the chyme of the colon was reduced by 50.8% compared with those in the HFD control group (*p* < 0.05) and it did not differ from those in the SCD control group.

APN activity in the colon chyme was increased in the HFD control group compared to the SCD control, IGT and T2D groups (by 2.0, 1.7 and 2.0 times, *p* < 0.05, *p* > 0.05 and *p* < 0.05, respectively) (Figure 10B).

### 3.7. Relationship between the AP Activity in the Chyme of the Colon and its Activity in the Mucosa of the Duodenum

Data on rats of different groups were used to analyze the relationship between AP activity in the chyme of the colon and its activity in the duodenal mucosa. A positive linear correlation exists between the alkaline phosphatase activities in the chyme of the colon and in the mucosa of the duodenum (coefficient of correlation r = 0.796) (Figure 11). This may indicate that the change in the activity of alkaline phosphatase in the chyme of the colon may be associated with the secretion of duodenal alkaline phosphatase into the intestinal lumen.

## 4. Discussion

In this work, we investigated for the first time the ability of the small intestine in growing rats to absorb glucose under in vivo conditions that are as close to natural as possible (in the absence of anesthesia and surgical trauma) during the development of type 2 diabetes and impaired glucose tolerance. The use of such experimental conditions is very important both in terms of assessing the actual levels of glucose absorption in the small intestine and in terms of adequately assessing the responses of intestinal glucose transporters SGLT1 and GLUT2 to T2D. This is justified by the fact that the anesthesia and surgical trauma inherent to and frequently used in acute in vivo experiments significantly reduces nutrient absorption in the small intestine, which can significantly distort the relative contribution of glucose transporters SGLT1 and GLUT2 to the resulting glucose absorption [40,41,42].

In studies of glucose absorption in the small intestine in vivo in the absence of anesthesia and surgical trauma, several variants of the approaches were used [40,41,42]. In this work, we used a new methodology based on our proven thesis about the existence of a direct relationship between the rate of free (voluntary) consumption of a concentrated glucose solution in rats that had previously fasted for 18–20 h and the ability to absorb glucose in the small intestine of rats [33,34].

Our results showed that 3 weeks after the first administration of streptozotocin (the inducer of type 2 diabetes), over the next 8 weeks, there was a more rapid increase in the glucose absorption capacity of the small intestine in the T2D group compared to the HFD control. This increase in glucose absorption was accompanied by a trend towards an increase (compared to HFD control) in the number of enterocytes on the villi of the jejunum and ileum in the T2D group (by 15% and 20%, respectively, 0.05 < *p* < 0.1), which was determined at the end of the experiments. These facts are in good agreement with other previously obtained data on T2D models in rats, including models on young animals [3,4,7,8,9,13]. At the same time, the immunohistochemical analysis of the tissue samples of the jejunum showed no changes in the content of glucose transporters SGLT1 and GLUT2 in the brush border membrane of enterocytes in the T2D group compared to the HFD control, which was not consistent with many of the results of other researchers. This can be due to a variety of reasons. In particular, this may be due to the fact that for the determination of the content of glucose transporters in the apical membrane of enterocytes in our experiments, we used a semi-quantitative approach, including an immunohistochemical method and confocal microscopy, which limits the possibility of evaluating small changes in the analyzed parameter. This could be due to the fact that type 2 diabetes in rats was less severe in these experiments than in the work of other authors and in our previous study in adult rats [3,4,7,8,9,13]. Thus, the area under the glucose curve in the OGTT test in this work was 22.8–24.6 mmol/L · hour, while in our previous study it was 29.1–57.7 mmol/L·h. In addition, due to the duration of the experiment (9 weeks after the induction of T2D) and the young age of the rats, glucose absorption might adapt to T2D according to the following scheme: from an increase in glucose absorption at the enterocyte level during a short period in a state of type 2 diabetes (for example, up to 5 weeks after its induction) until absorption increases due to an increase in the number of enterocytes on the villi. It is also possible that a certain contribution to the increased ability to absorb glucose in the T2D group may be made by the paracellular mechanism of glucose transport, since a number of studies have reported an increase in the permeability of the small intestine in type 1 and type 2 diabetes [43,44]. In addition, it should be noted that in the literature there is a similar example of the adaptation of glucose absorption in the small intestine to T2D, for example, in the Zucker diabetic fatty rats [8]. It was found that the increased absorption of glucose in the small intestine in Zucker diabetic fatty rats was caused by an increase in the intestinal mucosa mass, while the activity of glucose transporters SGLT1 and GLUT2 in the enterocytes did not change.

It is also interesting to remark a decrease (compared to SCD and T2D groups) in the number of goblet cells on the villi of the jejunum and ileum in the IGT group, which may indicate a decrease in mucus production and, as a consequence, a weakening of the protective function of the intestine in IGT rats.

In the present study, we also identified the specific effects of IGT and T2D on the activities of key intestinal digestive enzymes. Thus, in both the IGT and T2D groups, there was a significant decrease in the activity of glucoamylase in the duodenum compared with the controls, SCD and HFD. Considering that the activity of intestinal glucoamylase, determined in vitro, largely depends on the presence of the pancreatic α-amylase adsorbed on the intestinal surface, it may be assumed that IGT and T2D can reduce the adsorption properties of the pre-membrane layer for α-amylase. In addition, in contrast to the ITG group, the T2D group also showed a decrease in maltase activity in the distal jejunum compared to the SCD and T2D groups. What molecular mechanism underlies this phenomenon remains to be seen. At the same time, there is a reason to consider this reaction to be a protective one, since under these conditions, the absorption of glucose formed during the hydrolysis of its biopolymers can be limited, which, as a result, can help to reduce hyperglycemia during the digestion under conditions of type 2 diabetes.

In our experiments, the IGT and T2D groups also showed an increase in AP activity in the duodenal and colon mucosa compared to the HFD control. At the same time, in the IGT group, but not in the T2D group, AP activity also increased in the mucosa of the distal jejunum (compared to the SCD and HFD controls) and ileum (compared to the SCD control). If we bear in mind that AP is involved in the regulation of fat absorption in the small intestine [23,45], then, in the case of feeding on a high-fat diet, it is impossible to increase the AP activity in the mucosa of the small intestine in T2D rats, as it happens in IGT rats, which may promote higher intestinal fat absorption and obesity. In particular, our data on higher weight gain in T2D rats than in IGT rats during the course of the experiments are in good agreement with this conclusion.

AP is known to play an active part in the detoxification of this toxin [23,24,25]. In this work, we showed the presence of a close correlation between the activity of AP in the duodenal mucosa and the level of its activity in the contents of the colon. This confirms that alkaline phosphatase, which is produced in large quantities in the duodenum, moves to the large intestine and is involved there in the detoxification of bacterial LPS. However, the activity of AP in the contents of the colon in our experiments with the T2D group was noticeably lower than in the HFD and IGT groups. This is in good agreement with other data in the literature [26,27,28] that show a decreased level of AP in the chyme of the colon in humans with type 2 diabetes and in animal models involving type 2 diabetes. This fact suggests an increased risk of developing LPS-induced inflammation in the colon with T2D.

Regarding the other important intestinal enzyme, APN, in our experiments in the T2D group, there was a trend towards an increase in its activity in the jejunum (distal region) and ileum compared to the ITG group and HFD control. In the ITG group, the activity of APN increased significantly in the mucosa of the colon. In addition, in both groups (IGT and T2D), APN activity in the colon content was reduced compared to the HFD group. The APN, along with participation in digestion (the hydrolysis of food oligopeptides), also performs other important functions. In particular, it is involved in cholesterol transport, immune responses and the degradation of biologically active peptides [23,29,30,31,32]. Thus, an increase in the activity of this enzyme in the intestinal mucosa or its decrease in the chyme of the colon can have adverse effects on the health of the body. As can be seen from our results, the effects of T2D and NTG on the activity of APN in the intestine also differ.

## 5. Conclusions

In summary, the presented data show that the adaptations of intestinal digestive enzymes and glucose absorption systems in the small intestine in growing animals with IGT and T2D have both common features and specific differences. The most significant differences are the increased capability of the small intestine to absorb glucose and no increase in the activity of alkaline phosphatase in the mucosa of the small intestine and in the contents of the colon with T2D. The increased absorption of glucose in the small intestine is at least in part associated with an increase in the number of enterocytes on the villi. Further research is needed to further elucidate the molecular mechanisms underlying these changes.

## Figures and Tables

**Figure 1 nutrients-14-00385-f001:**
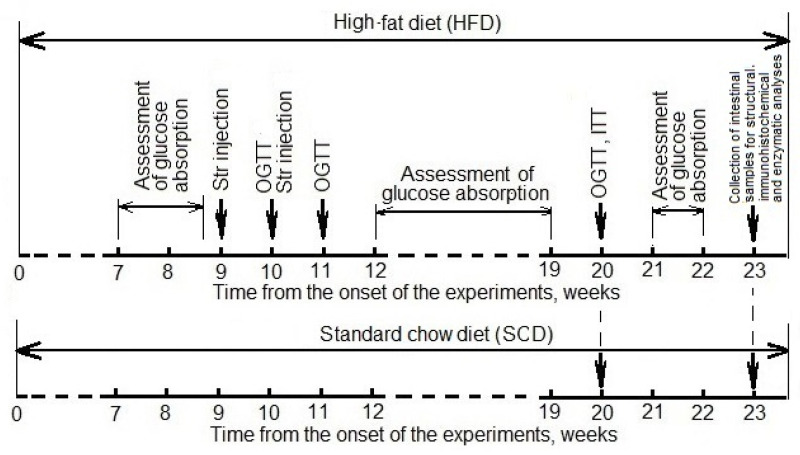
A design of the experiments. Str—streptozotocin; OGTT—oral glucose tolerance test, ITT—insulin tolerance test.

**Figure 2 nutrients-14-00385-f002:**
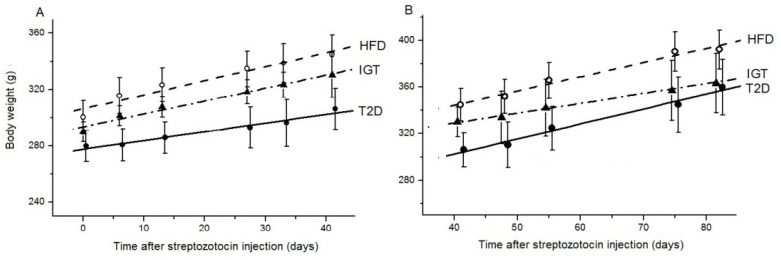
Changes in the body weight of rats in the groups with type 2 diabetes mellitus (T2D) and impaired glucose tolerance (IGT) after the administration of streptozotocin, and in the control group (HFD) after the administration of its solvent. (**A**) In the first 40 days; (**B**) in the second 40 days. (**A,B**) The T2D group (*n* = 4), the IGT group (*n* = 6) and the HFD group (*n* = 6). Results are expressed as means ± the standard error of the mean (SEM). Straight lines are the regression lines.

**Figure 3 nutrients-14-00385-f003:**
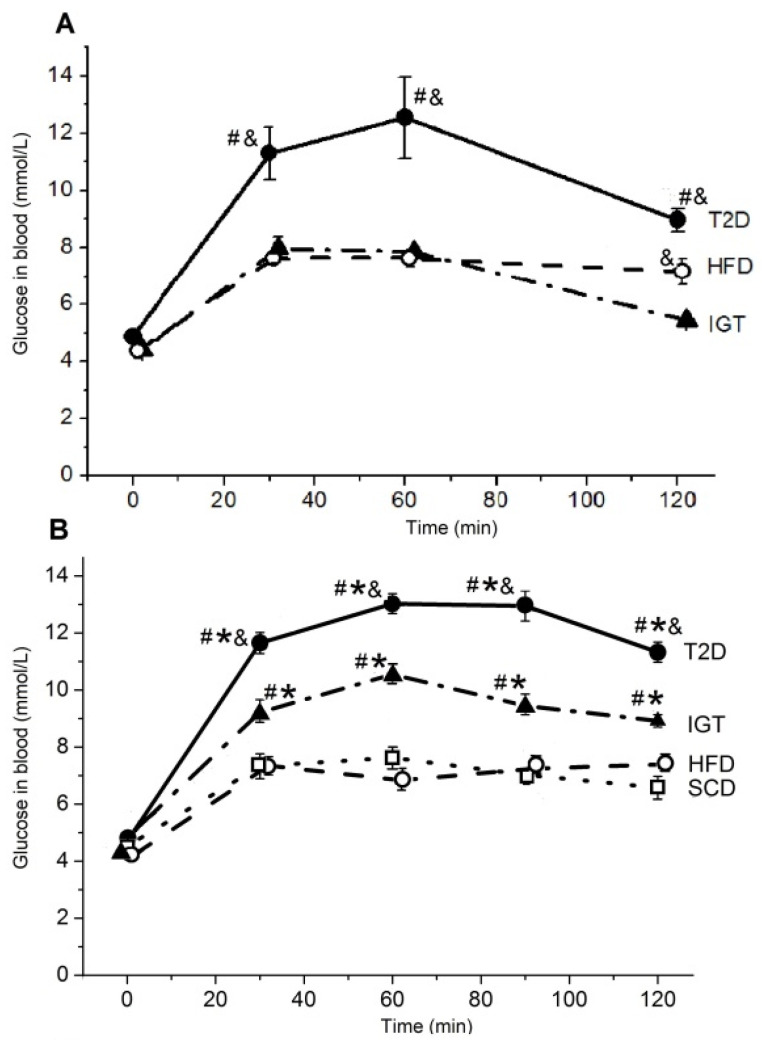
Results of the oral glucose tolerance test (OGTT) and the insulin tolerance test (ITT) in the rats in the groups with type 2 diabetes (T2D), with impaired glucose tolerance (IGT) and in control groups: high-fat diet (HFD) and standard chow diet (SCD). (**A**) The OGTT was performed 2 weeks after the administration of streptozotocin or its solvent. (**B**) The OGTT was performed 10 weeks after the administration of streptozotocin or its solvent to the rats. (**C**) The ITT was performed 11 weeks after the administration of streptozotocin or its solvent. (**A**–**C**) The T2D group (*n* = 4), the IGT group (*n* = 6), the HFD group (*n* = 6) and the SCD group (*n* = 6). Results are expressed as means ± the standard error of the mean (SEM). Two-way ANOVA: * *p* < 0.05 compared with the SCD group, # *p* < 0.05 compared with the HFD group, & *p* < 0.05 compared with the IGT group.

**Figure 4 nutrients-14-00385-f004:**
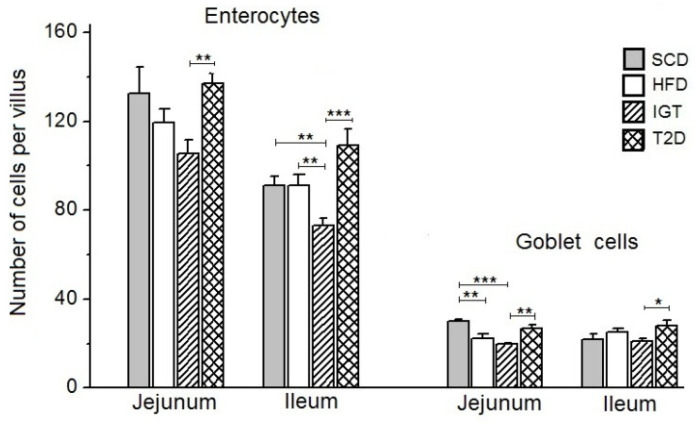
The number of enterocytes and goblet cells in the villi of the jejunum and ileum in the rats with type 2 diabetes (T2D), with impaired glucose tolerance (IGT) and in control groups: high-fat diet (HFD) and standard chow diet (SCD). The T2D group (*n* = 4), the IGT group (*n* = 6), the HFD group (*n* = 6) and the SCD group (*n* = 6). Results are expressed as means ± the standard error of the mean (SEM). Student *t*-test: * *p* < 0.05, ** *p* < 0.02 and *** *p* < 0.0027.

**Figure 5 nutrients-14-00385-f005:**
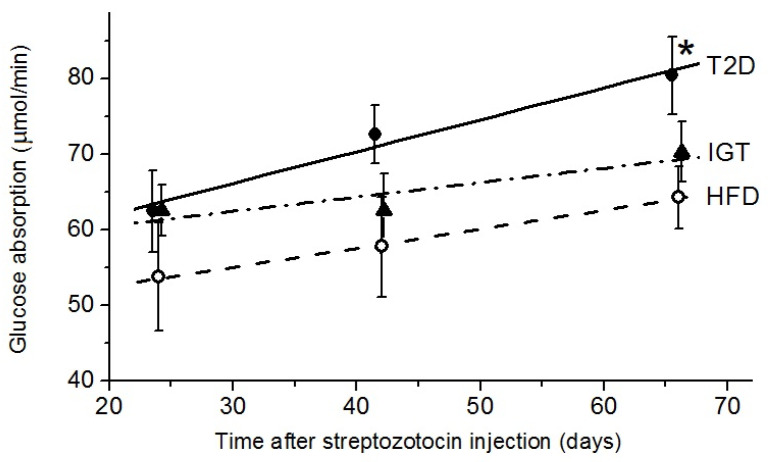
The maximal rate of the absorption of glucose (μmol/min) in the small intestine of rats with type 2 diabetes (T2D), with impaired glucose tolerance (IGT) and in the HFD control group (a high-fat diet) at different periods of the experiment. The T2D group (*n* = 4), the IGT group (*n* = 6) and the HFD group (*n* = 6). Results are expressed as means ± the standard error of the mean (SEM). Straight lines are the regression lines. Student *t*-test: * *p* < 0.05 compared with the HFD group.

**Figure 6 nutrients-14-00385-f006:**
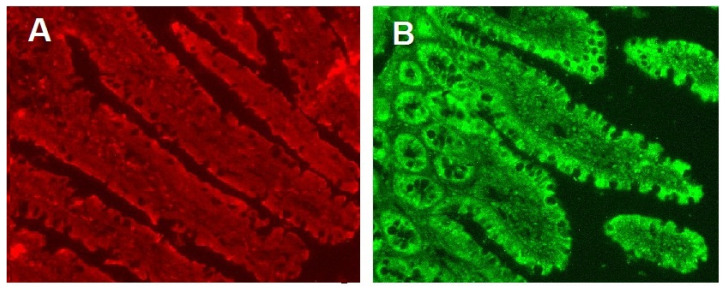
The results of the immunohistochemical analysis of transporters SGLT1 and GLUT2 in rat jejunal enterocytes. (**A**) SGLT1, (**B**) GLUT2. Since a quantitative assessment of the content of these transporters in the apical membrane of enterocytes did not reveal differences in type 2 diabetes and HFD control, images for each transporter are presented in one copy.

**Figure 7 nutrients-14-00385-f007:**
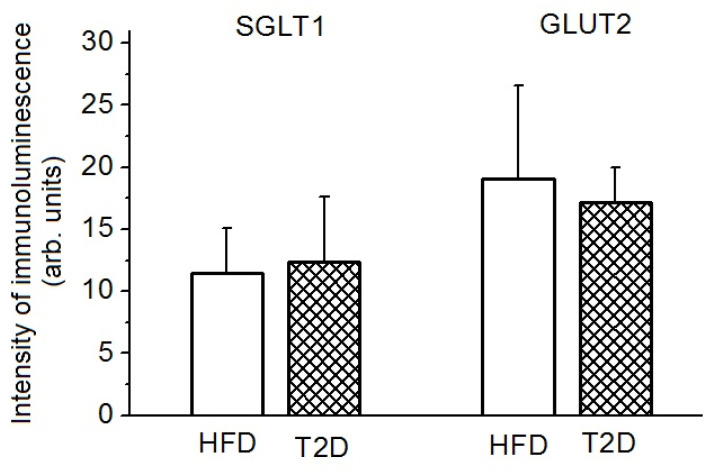
The content of transporters SGLT1 and GLUT2 in the brush border membrane of jejunal enterocytes in the T2D and HFD groups. The T2D group (*n* = 4) and the HFD group (*n* = 6). Results are expressed as means ± the standard error of the mean (SEM).

**Figure 8 nutrients-14-00385-f008:**
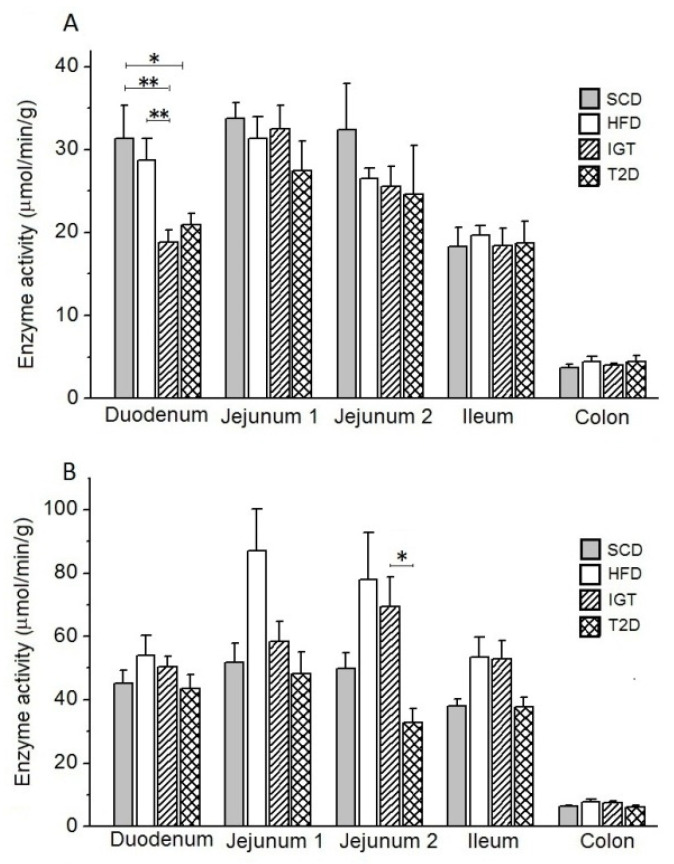
Activities of glucoamylase and maltase in different parts of the intestine in rats of 23 weeks old with type 2 diabetes (T2D), with impaired glucose tolerance (IGT) and in control groups: high-fat diet (HFD) and standard chow diet (SCD). (**A**) Glucoamylase activity; (**B**) maltase activity. (**A**,**B**) Jejunum 1 and 2 are proximal and distal halves of the jejunum, respectively. The T2D group (*n* = 4), the IGT group (*n* = 6), the HFD group (*n* = 6) and the SCD group (*n* = 6). Results are expressed as means ± the standard error of the mean (SEM). Student t-test: * *p* < 0.05 and ** *p* < 0.0027.

**Figure 9 nutrients-14-00385-f009:**
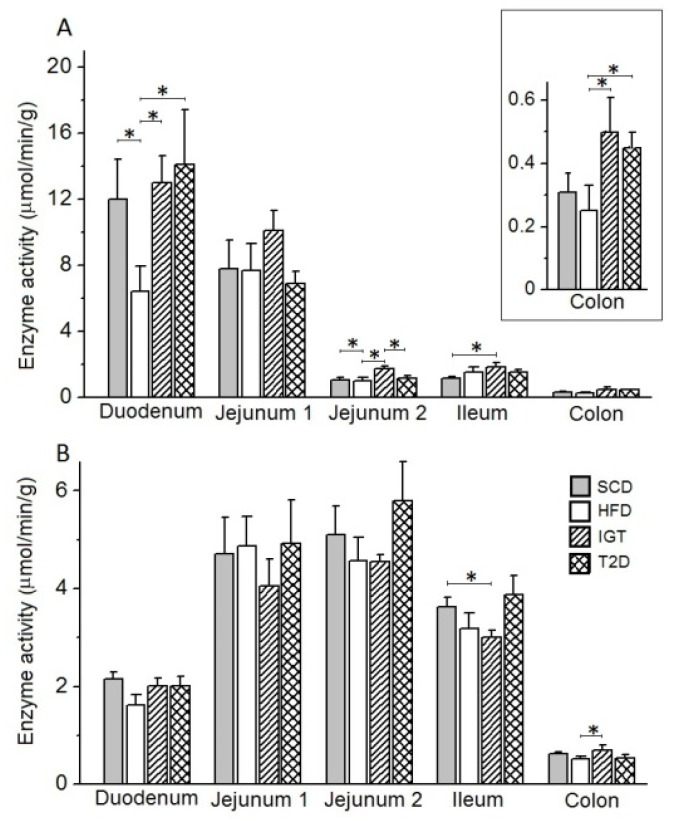
Activities of alkaline phosphatase and aminopeptidase N in the mucosa of various parts of the intestine in rats with type 2 diabetes (T2D), with impaired glucose tolerance (IGT) and in control groups: high-fat diet (HFD) and standard chow diet (SCD). (**A**,**B**) Jejunum 1 and 2 are proximal and distal halves of the jejunum, respectively. The T2D group (*n* = 4), the IGT group (*n* = 6), the HFD group (*n* = 6) and the SCD group (*n* = 6). Results are expressed as means ± the standard error of the mean (SEM). Student *t*-test: * *p* < 0.05.

**Figure 10 nutrients-14-00385-f010:**
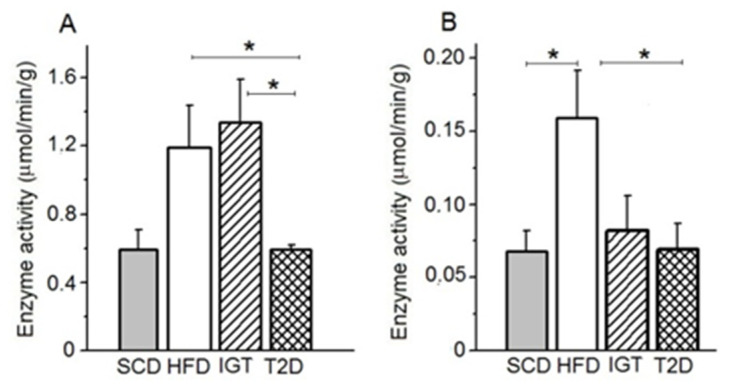
Activities of alkaline phosphatase and aminopeptidase N in the chime of the colon in rats with type 2 diabetes (T2D), with impaired glucose tolerance (IGT) and in control groups: high-fat diet (HFD) and standard chow diet (SCD). (**A**) Alkaline phosphatase activity; (**B**) aminopeptidase N activity. (**A**,**B**) The T2D group (*n* = 4), the IGT group (*n* = 6), the HFD group (*n* = 6) and the SCD group (*n* = 6). Student *t*-test: * *p* < 0.05.

**Figure 11 nutrients-14-00385-f011:**
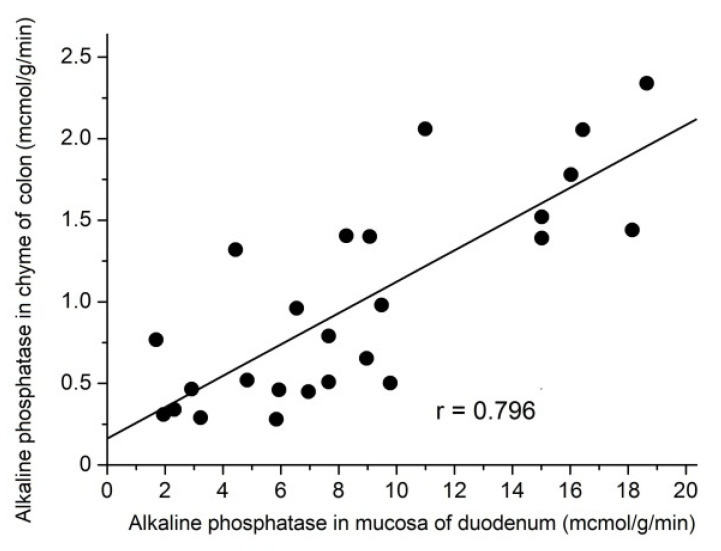
Linear correlation between alkaline phosphatase activities in jejunal mucosa and in the colon chyme. Coefficient of correlation r = 0.796 (*n* = 25). The correlation was analyzed using Origin 7.0 (OriginLab Corporation, Northampton, MA, USA).

## Data Availability

Data available in a publicly accessible repository.

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
