# Peer review of "Effect of Type 2 Diabetes and Impaired Glucose Tolerance on Digestive Enzymes and Glucose Absorption in the Small Intestine of Young Rats"

_nutrients, 2022, doi:10.3390/nu14020385_

Round 1
Reviewer 1 Report
This manuscript summarizes results of an experimental study in young rats on a high-fat diet following pathophysiological changes (glucose transport, gut morphological and immunohistochemical characteristics, activity of some intestinal digestive enzymes) after administration of streptozotocin to induce a (pre-)diabetic situation. Unfortunately, data presentation is in many parts imprecise, difficult to follow and confusing.
General comments
- The overall goal of the study remains unclear. Why rats were kept on a HDF diet? Where is the link to pathophysiological changes after streptozotocin infusion? If it was planned to compare nutritive effects (HFD vs. SCD) in this diabetes model, why SCD rats were not treated with streptozotocin infusion? The authors report that they differentiate between IGT and T2D – why? And was this differentiation simply „by chance“ (not originally planned) due to nonuniform pathophysiological reactions following streptozotocin (lines 142-150)? In this context it is interesting (and incomprehensible) that some animals of the experimental groups received a second streptozotocin dose because „signs of type 2 diabetes or impaired glucose tolerance“ (??) could not be detected.
Independent of these obvious inconsistencies it is, indeed, questionable whether this study provides novel information. As mentioned by the authors themselves, many experimental studies have already been published presenting diabetes-specific alterations in glucose transport etc.
Specific
- In the present form this abstract makes no real sense. Important information is not given: rationale, number of rats, composition of diets, results (!!) etc. Should be completely reworded.
- Much too long due to several parts not related to the topic. More important, the rationale to perform this experimental study is not worked out.
- Study design/methods. Table 1 is confusing, especially the overlapping time table. Alternatively, a figure (time line) with additional text should be more clear and informative.
Some information is lacking: Composition of SCD? Why 6/7 animals per group? All micronutrients supplied in both diets? Reliability, sensitivity etc. of the analytical methods used?
- Figure 1: What about body weight in the SCD rats? Figure 2A: Why data obtained with SCD rats are not shown? Figures 5/6: Why images/data for SCD rats are not shown?
The accompanying text is too long; information obviously provided in the figures should not be „repeated“.
- Again, it remains unclear whether the authors used a HFD for their experiment (indeed, not mentioned in the first paragraph!). In any case, the authors should comment on this point.
To mention that „..a new methodology based on our proven thesis..“ (line 410) without discussing whether this method is „better“ than the old one makes not really sense.
Lines 426/427: „..less pronounced severity of type 2 diabetes in our experiments than in the (previous) experiments..“: What is meant? What is the consequence – comparisons are not allowed?
Line 444: „Our hypothesis…“ which one?
Lines 448f: Repetitions of results should be avoided. Linke to the respective figures/tables is a clear alternative.
- Again, the „nutritive approach“ of this study is not mentioned at all!
Author Response
Please, see attached file.

Reviewer 2 Report
Dear Authors,
The manuscript "Effect of Type 2 Diabetes Caused by a High-Fat Diet/Low-Dose Streptozotocin on Digestive Enzymes and Glucose Absorption System in the Small Intestine of Young Rats" focus on the effect of a high fat diet/Low Dose Steptozocin on digestive enzynesis and glucos absorption in a tyoe 2 diabetes model. The manuscript is interesting and clinical relevant.
The introduction can be improved. From line 52 to 58, the information is confusing.
The experimental design is appropriate to test the hypothesis and the manuscript’s results reproducible based on the the methods described.
Author Response
Please, see the attached file.

Round 2
Reviewer 1 Report
Thanks for the efforts to consider the criticisms raised in the revised manuscript. Indeed, some (but not all) points are now more clear for the reader. Moreover, not all comments given in the accompanying letter are adequately included in the revised version of the manuscript.
- To clarify the goal and to hinder misunderstandings (as mentioned in the letter: HFD was only a tool to produce T2D!) it might be wise to reword the title: Effect of Type 2 Diabetes and IGT on Digestive Enzymes and Glucose Absorption in the Small Intestine of Young Rats.
- Still not acceptable! It is simply not true that the word limitation hinders the inclusion of important information. An abstract should provide a brief goal, methods (T2D model, number of animals), objective results (some figures) and a conclusion. All other text can be deleted.
- Lines 74f and other parts of the manuscript: If it was a „goal“ to „compare“ two groups of animals (T2D and IGT) then the whole corresponing text in the manuscript should be written accordingly. If it was „by chance“ that animals did not react as expected this fact should be discussed as a limitation of the study design.
- Lines 90f: It should be clearly reported that the use of a HFD is part of the model to induce T2D. Following this idea it is still not to understand why a „control“ group consuming a SCD is part of the study design (no objective results reported for that group!).
- Table 1 is almost unchanged!? Why not including a figure as suggested?
- The comments in the letter ranking the methodology used as „semiquantitative etc.“ must be included in the manuscript (Discussion – limitations).
Author Response
See attached file.

This manuscript is a resubmission of an earlier submission. The following is a list of the peer review reports and author responses from that submission.
Round 1
Reviewer 1 Report
The author reported that it is an increase in glucose absorption in the small intestine and a decrease in alkaline phosphatase activity in the colon contents. 
Studies using IAP knockout mice have shown that IAP enzymes are involved in the rate-limiting step of fat absorption. Studies using IAP knockout mice also suggest that the enzyme is involved in the rate-limiting step of fat absorption and that the absence of the enzyme may cause metabolic abnormalities (Nakano T, et al: G1439-G1449, 2007:Narisawa S, et al. Mol. Cell Biol 23: 7525-7530, 2003).
How are glucose and fat absorption and alkaline phosphatase activity in the colon contents involved?
Please mention fat absorption.
Author Response
Please, see the attachment.

Reviewer 2 Report
Gromova et al., have investigated the effect of type 2 diabetes caused by HFD/low dose Streptozotocin on digestive enzymes and glucose absorption in the small intestine. Whilst the topic is of relevance, in this format, it is hard to understand the text, and identify what the key results are and how to interpret them. This article was hard to read due to english being a second language and because of this, the article was not clear as to the experimental intent and how the results confirmed their hypothesis.
Major concerns are:
- the methods are inadequate to be able to understand how experiments were conducted. There were no catalogue numbers or company details about any reagents, methods are missing and the readers have no idea what samples were taken and what was measured in them from the rats. Especially for intestinal permeability and glucose absorption, the two key methods the whole paper is based on.
- The method for the ITT included fasting overnight which reduces blood glucose before starting. Insulin administration then caused it to drop to dangerously low levels that were close to toxic.
- Statistical tests used for the OGTT and ITT should be 2 way ANOVA not t tests. No stars are presented on graphs and the text refers to all experiments being different.
- The experiment for mannitol absorption only presents results as text, not graphical, which is strange as it is a key experiment.
- It would be great to see representative images for the GLUT2 expression as well as the quantification.
- It is recommended to the authors in the results section to state what each result means to the overall aim or hypothesis. It was not clear and made the article hard to understand.
Round 2
Reviewer 2 Report
the methods are inadequate to be able to understand how experiments were conducted. There were no catalogue numbers or company details about any reagents, methods are missing and the readers have no idea what samples were taken and what was measured in them from the rats. Especially for intestinal permeability and glucose absorption, the two key methods the whole paper is based on. Answer: 1. The comment is accepted. The Materials and Methods section was redesigned: a table on the design of experiments and brief descriptions were added for the method for assessing glucose absorption in the small intestine and for the method for determining the content of glucose transporters in enterocytes (see the updated text of the manuscript).
Upon revision, the table detailing the timeline of experiments in the rats was a great way to understand the order of experiments, however, some of the methodology still does not explain what endpoint is measured. The problems still are:
- The glucose absorption assay does not measure absorption adequately. It is more a glucose preference test or glucose consumption test. If measuring glucose absorption then PET imaging or glucose tracers should be performed. All we can conclude from the experiment conducted is that the rats consume glucose at different rates. The equation that calculates the umol/min needs to be described in the methods. The authors state "This technique bases on the fact that rats, after fasting for 18 - 20 hours, consume concentrated glucose solution at almost constant average rate during several hours due to so 103 called “ileal brake” – the mechanism of control of the stomach emptying and intestinal 104 motility according to the rate of nutrients absorption in the small intestine [35]. However, the small intestine villi length changes with HFD and T2D is also known to change the ileal architecture which would mean the equation assuming that the intestinal environment is the same can not be applied to your experimental groups.
- The mannitol experiment is still not describing where and how the mannitol is measured. Authors state "which was determined by the absorption of mannitol transferred through enterocytes by passive transport. Mannitol absorption was evaluated using its 10% solution of this substrate by rats". Is the intestinal segment removed and absorption removed in intestinal pieces? More description is needed.
- Then, samples of tissues of the middle part of the jejunum 143 and ileum were taken for the analysis of morphometric parameters and the content of 144 SGLT1 and GLUT2 transporters in enterocytes using the immunohistochemical method. Please add in "this method is described below in section 2.7.
Figure 4. Glucose absorption. This experiment is presented as multiple measurements in the same rats over time. Therefore a 2 way ANOVA should be employed not a t-test.
Could the authors comment as to why they did not see any differences in SGLT1 protein expression when it has been shown multiple times that STZ treatment in rats increases SGLT1 expression. How are the T2DM rats then increasing their glucose absorption if they do not have any differences in GLUT2 or SGLT1? Images must be shown in the Figure or supplementary figures. Protein expression or mRNA expression would help strengthen this experimental data that the transporter content is not different between groups. Also why was the IGT group omitted from this analysis?
In this work, we at the first time investigated the absorption of glucose in the small 379 intestine in growing rats under in vivo conditions that are as close to natural as possible 380 (in the absence of anesthesia and surgical trauma) during the development of type 2 di- 381 abetes and impaired glucose tolerance. The authors state that their experiments provide a more realistic analysis of glucose utlilisation due by not using anaesthetic, however the methods the authors have tried to use to determine the glucose permeablility are not taking their experimental animals intestinal architecture into account. Perhaps the authors could perform ex vivo experiments on intestinal segments to perform more exact glucose permeability assays. The authors refer to two papers that describe their method, however both are from their laboratory and one is written in Russian. This method needs to be performed and described by other laboratories.